# Magnetical Manipulation of Hyperbolic Phonon Polaritons in Twisted Double-Layers of Molybdenum Trioxide

**DOI:** 10.3390/mi14030648

**Published:** 2023-03-13

**Authors:** Hongjing Li, Gaige Zheng

**Affiliations:** 1School of Electronics Engineering, Nanjing Xiaozhuang University, Nanjing 211171, China; 2Jiangsu Collaborative Innovation Center on Atmospheric Environment and Equipment Technology (CICAEET), Nanjing University of Information Science and Technology, Nanjing 210044, China

**Keywords:** phonon polaritons, van der Waals crystals, Indium arsenide

## Abstract

Controlling the twist angle between double stacked van der Waals (vdW) crystals holds great promise for nanoscale light compression and manipulation in the mid-infrared (MIR) range. A lithography-free geometry has been proposed to mediate the coupling of phonon polaritons (PhPs) in double-layers of vdW α-$MoO_{3}$. The anisotropic hyperbolic phonon polaritons (AHPhPs) are further hybridized by the anisotropic substrate environment of magneto-optic indium arsenide (InAs). The AHPhPs can be tuned by twisting the angle between the optical axes of the two separated layers and realize a topological transition from open to closed dispersion contours. Moreover, in the presence of external magnetic field, an alteration of the hybridization of PhPs will be met, which enable an efficient way for the control of light-matter interaction at nanoscale in the MIR region.

## 1. Introduction

Recently, there has been growing interest in the study of PhPs, where the light is coupled to an optical phonon mode in polar crystals instead of a plasmon in metal [1,2,3,4]. PhPs can be excited in the Reststrahlen band (RB) between the longitudinal optical (LO) and transverse optical (TO) frequencies of polar dielectrics [5,6]. Owing to the significantly longer lifetimes of optical phonons compared to plasmons, PhPs have been widely used for engine absorption, controlling thermal emission, and achieving sub-diffraction optical confinement at longer wavelengths.

Strongly anisotropic materials can support hyperbolic phonon polaritons (HPhPs) that exhibit a hyperbolic dispersion whose permittivity tensor posses both positive and negative principal components [2,7,8,9]. There has been significant progress in natural vdW crystal and heterostructures, which are characterized by an anisotropic polaritonic response, leading to elliptical, hyperbolic, or biaxial polaritonic dispersions [10,11]. Different polaritonic modes in vdW materials have been discovered, such as plasmon polaritons in graphene, exciton polaritons in molybdenum diselenide ($MoSe_{2}$), phonon polaritons in hexagonal boron nitride (hBN), and $MoO_{3}$ [12,13,14,15,16]. Among these structures, PhPs with hyperbolicity in α-$MoO_{3}$ are of particular interest, not only due to the strong field confinement, low losses, and long lifetimes, but also the natural in-plane anisotropic dispersion [7,9,17,18,19,20]. Recent discovery of AHPhPs holds great enhancement in the light–matter interaction and provides a promise platform in the fields of infrared optics and nanophotonics. However, a significant challenge is the lack of active tunability of these AHPhPs.

Here we propose and theoretically demonstrate a lithography-free approach for manipulating the in-plane AHPhPs in twisted double layers of α-$MoO_{3}$ deposited on magneto-optic InAs substrate. The interaction of this device with MIR radiation is investigated using electromagnetic simulations that utilize the generalized 4 × 4 transfer matrix algorithm. The AHPhPs can be tuned by twisting the angle between the optical axes of the two separated layers and realize a topological transition from open to closed dispersion contours. More interestingly, an effective manipulation by incorporating with a static external magnetic field can be achieved. Beyond demonstrating a new way of controlling light in deeply subwavelength scales in the MIR, hybrid magneto-optic, and twisted α-$MoO_{3}$ structure reported here is of great potential for thermal radiation control and photodetectors owing to the low required magnetic fields for resonance tunability.

## 2. Materials and Methods

The schematic of the proposed structure is shown in Figure 1, the twist angle (θ) is illustrated in the Cartesian coordinate system. The optical response of α-$MoO_{3}$ is dominated by the phonon absorption rather than electronic transition, thus its dielectric tensor described by the following Lorentz model [18,21]:(1)$$\epsilon_{j}\left(\omega \right)=\epsilon_{\infty }^{j}\left(1+\frac{{\omega_{LO}^{j}}^{2}-{\omega_{TO}^{j}}^{2}}{{\omega_{TO}^{j}}^{2}-\omega^{2}-i\omega \gamma^{j}}\right)\;j=x,y,z$$
where $\epsilon_{j}$ denotes the principal components of the permittivity tensor. The $\epsilon_{\infty }^{j}$ is the high-frequency dielectric constant, the $\omega_{LO}^{j}$ and $\omega_{TO}^{j}$ refer to the LO and TO optical phonon frequencies, respectively. The parameter $\gamma^{j}$ is the broadening factor of the Lorentzian lineshape. The *x*, *y*, and *z* denote the three principal axes of the crystal, which correspond to the crystalline directions [100], [001], and [010] of the α-$MoO_{3}$, respectively. All used parameters are listed in Table 1 and the results are provided in Figure 2a,b. Due to the asymmetric lattice structure of α-$MoO_{3}$, PhPs in α-$MoO_{3}$ exhibit anisotropic propagation between the TO and LO phonon frequencies: hyperbolic in RB1 (545−851 cm${}^{-1}$) and RB2 (820−972 cm${}^{-1}$), and elliptical in RB3 (958−1004 cm${}^{-1}$). Here, we are particularly interested in RB2, where the permittivity components along the [100], [001], and [010] crystal directions satisfy $\epsilon_{x}<0$, $\epsilon_{y}>0$, and $\epsilon_{z}>0$, respectively. As a result, the inplane PhPs in natural α-$MoO_{3}$ exhibit a hyperbolic dispersion contour [22,23].

The degenerately doped InAs exhibits a Drude-like optical response with non-zero off-diagonal permittivity values in an applied magnetic field. We choose the magnetization parallel to the z axis, such that the permittivity tensor takes the form [24]:(2)$$\epsilon_{InAs}=\left[\begin{array}{ccc} \epsilon_{xx} & \epsilon_{xy} & 0\\ \epsilon_{yx} & \epsilon_{yy} & 0\\ 0 & 0 & \epsilon_{zz} \end{array}\right]$$
(3)$$\epsilon_{xx}=\epsilon_{yy}=\epsilon_{\infty }-\frac{\omega_{p}^{2}\left(\omega +i\Gamma \right)}{\omega ({\left(\omega +i\Gamma \right)}^{2}-\omega_{c}^{2})}$$
(4)$$\epsilon_{xy}=-\epsilon_{yx}=i\frac{\omega_{p}^{2}\omega_{c}}{\omega ({\left(\omega +i\Gamma \right)}^{2}-\omega_{c}^{2})}$$
(5)$$\epsilon_{zz}=\epsilon_{\infty }-\frac{\omega_{p}^{2}}{\omega (\omega +i\Gamma )}$$

The electromagnetic waves with transverse magnetic (TM) polarization $\left(E_{x},H_{y},E_{z}\right)$ launch into the structure with an incident angle. $\omega_{p}=\sqrt{\frac{ne^{2}}{m^{*}\epsilon_{0}}}$ is the plasma frequency, where *n*, *e*, and $m^{*}$ are the free electron density, charge, and effective electron mass, respectively. $\omega_{c}=\frac{eB}{m^{*}}$ denotes the cyclotron frequency, which is dependent on the external magnetic field *B*. The relaxation rate Γ = 4.5 THz, a high-frequency limit dielectric constant $\epsilon_{\infty }=12.3$, an electron density $n=1.5\times 10^{18}$ cm${}^{-3}$, and an effective electron mass $m^{*}=0.033m_{e}$, where $m_{e}=9.109\times 10^{-31}$ kg.

## 3. Results and Discussions

Due to the TM-polarized evanescent waves occupies the main contribution in the manipulation of AHPhPs, an intensity of the imaginary part of the Fresnel reflection coefficient $Im(r_{pp})$ will be investigated. The dispersion relations of the above-mentioned heterostructures are calculated by using the 4 × 4 matrix [25,26,27]. As shown in Figure 3, the hybridized PhPs dispersion of the α-$MoO_{3}$/α-$MoO_{3}$/InAs structure along its main crystallographic directions are calculated with $\theta =0^{\circ }$, $\theta =45^{\circ }$, and $\theta =90^{\circ }$, respectively. The thickness of the α-$MoO_{3}$ layer is 200 nm and the quantum confinement effects are ignored. Double hybridization PhPs dispersion modes at $\theta =0^{\circ }$ are gathering inside RB2 and RB3. Similarly, double PhPs dispersion modes of the structure at $\theta =90^{\circ }$ occur at RB1 and RB3. Interestingly, the PhPs dispersion modes at $\theta =45^{\circ }$ occur simultaneously within its hyperbolic bands, which can be attributed to the overall consideration of the rotated permittivity tensor between different crystalline directions.

Moreover, through calculating the angular-dependent dispersion relations, we could construct the isofrequency surfaces of the AHPhPs, as shown in Figure 4. The AHPhPs dispersion relations for twisted hyperbolic system are well predicted by the reflection coefficients. When $\theta =0^{\circ }$, one notices that the hybrid AHPhPs associated with individual α-$MoO_{3}$ can interact with each other, resulting in three dispersion curves in a higher wave vector region and a smaller wave vector region. As θ increases, the contour of the reflection coefficient exhibits a range of patterns from an open hyperbola-like pattern to a closed circumference-like pattern, experiencing a topological transition. The change of the hybrid HPhPs propagation is attributed to polaritonic coupling between individual α-$MoO_{3}$ in the twisted stack.

As the energy of hybrid coupling in our design is also determined by the property of substrate. If n-type-doped magneto-optic InAs is used as the substrate, active control of AHPhPs with a static magnetic field can be realized. In order to understand how the optical properties of the hybrid states are affected by the variation of an external magnetic field, $Im(r_{pp})$ as a function of in-plane momenta $k_{x}$ and $k_{y}$ under an external magnetic fields of 2T is shown in Figure 5. The result suggests that the in-plane propagation of HPhPs in the 200 nm α-$MoO_{3}$/200 nm α-$MoO_{3}$/InAs heterostructure. As the permittivity of InAs substrate can be easily modified by magnetic fields, active control of the coupling between the double α-$MoO_{3}$ layers can be realized. Therefore, the shape of the hyperboloid varies with varying external magnetic fields.

Furthermore, the hybridization dispersion modes of the α-$MoO_{3}$/α-$MoO_{3}$/InAs structure varying with the wavevector components $k_{x}$ and $k_{y}$ at the wavenumbers 600 cm${}^{-1}$ and 840 cm${}^{-1}$ are also calculated in Figure 6a–f to illustrate the differences. The analytical expression for the dispersion relations can be calculated approximately [28,29]:(6)$$\frac{k_{\rho }}{k_{0}}\left(\omega \right)=\frac{\psi }{k_{0}d}\left[\arctan\left(\frac{\psi }{\epsilon_{z}}\right)+\arctan\left(\frac{\psi \epsilon_{InAs}}{\epsilon_{z}}\right)+m\pi \right]\;m=0,1,2\cdots$$
(7)$$\psi =i\sqrt{\frac{\epsilon_{z}}{\epsilon_{x}\cos{\left(\theta \right)}^{2}+\epsilon_{y}\sin{\left(\theta \right)}^{2}}}$$
Here, $\epsilon_{InAs}$ is the permittivity of InAs, *d* denotes the α-$MoO_{3}$ flake thickness, $\epsilon_{z}$ is the α-$MoO_{3}$ permittivity along the [010] crystalline direction, and *m* is an integer.

The wavefront of phonon polaritons highlights a topological transition. In particular, in Figure 6a–c we show that for ω = 600 cm${}^{-1}$ the polaritonic mode launched at the region where real($\epsilon_{x}$) is positive and real($\epsilon_{y}$) is negative, has a typical hyperbolic behavior along *x*-direction. With the increasing of twist angle, both in-plane permittivity tensor elements will become negative, then the dispersion describes an ellipse, and the resulting SPhP can propagate along any direction in the surface plane [30,31]. In Figure 6d–f, we show that for ω = 840 cm${}^{-1}$ the polaritonic mode launched at the region where real($\epsilon_{x}$) is negative and real($\epsilon_{y}$) is positive, thus a hyperbolic phenomenon will appear along *y*-direction. The phonon polariton wavefront switches from hyperbolic to oval, appearing as slightly closed fringes (Figure 6f).

## 4. Conclusions

As the final remarks, the heterostructures of twisted α-$MoO_{3}$ on a n-type-doped magneto-optic InAs has been proposed to realize the active manipulation of the hyperbolic polaritons. The optical responses with MIR radiation is investigated using electromagnetic simulations based on the generalized 4 × 4 transfer matrix algorithm. We demonstrate the distinct wavelength tunability of the AHPhPs modes and the optical topologic transitions from open to closed iso-frequency contours as the increase in the twist angle. In the presence of external static magnetic field, the effective permittivity of InAs becomes affected and modified due to cyclotron frequency of electron, which leads the alteration in the dispersion relation of hybrid AHPhPs modes. The work presented here offers new prospects to the field of nanophotonics, exploring new light–matter interaction regimes of different optical modes and offering a platform for understanding, optimizing, and predicting new forms of polariton heterostructures.

## Figures and Tables

**Figure 1 micromachines-14-00648-f001:**
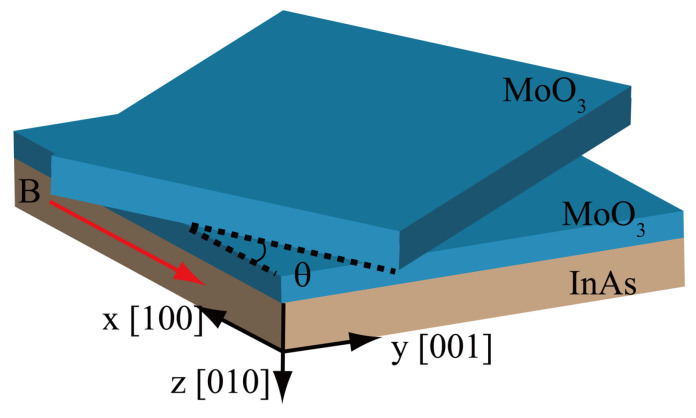
Schematic geometry of the twisted double-layer α-$MoO_{3}$ laminates deposited on magneto-optic InAs substrate.

**Figure 2 micromachines-14-00648-f002:**
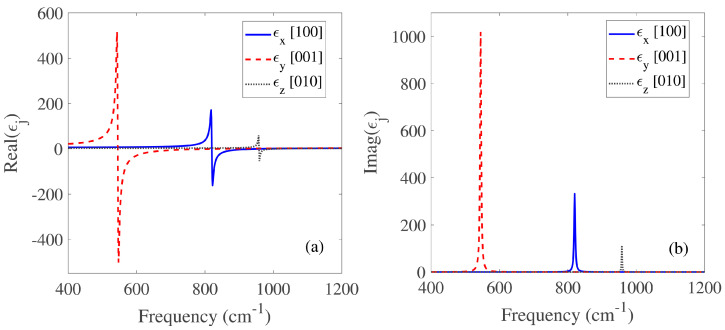
Real (**a**) and imaginary (**b**) parts of the dielectric function for α-$MoO_{3}$.

**Figure 3 micromachines-14-00648-f003:**
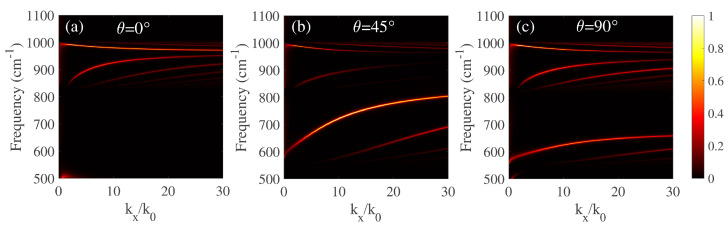
Dispersion curves under different twist angles. (**a**) $\theta =0^{\circ }$, (**b**) $\theta =45^{\circ }$, and (**c**) $\theta =90^{\circ }$. There is no external magnetic field in this case.

**Figure 4 micromachines-14-00648-f004:**
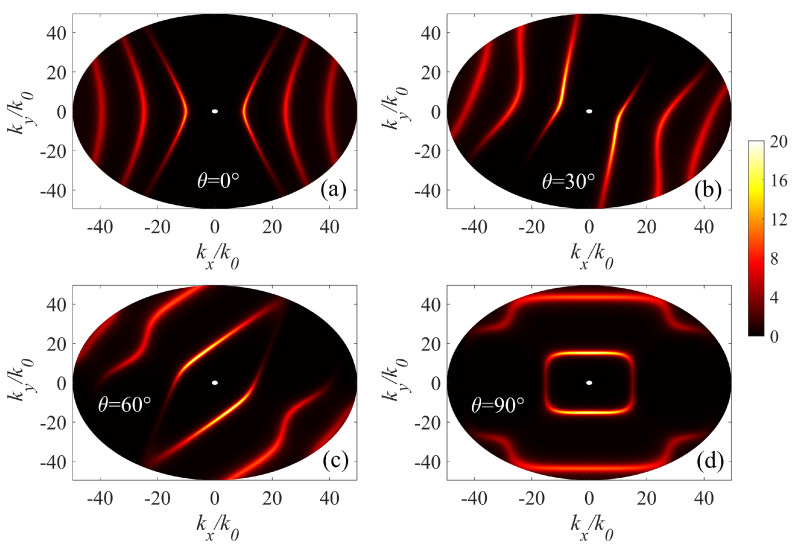
Contours of the imaginary part of the reflection coefficient ($Im(r_{pp})$) in the wave vector space at a frequency of 900 cm ${}^{-1}$ under different twist angles without any external magnetic field. (**a**) $\theta =0^{\circ }$, (**b**) $\theta =30^{\circ }$, (**c**) $\theta =60^{\circ }$, and (**d**) $\theta =90^{\circ }$. The structural parameters are the same as used in Figure 3.

**Figure 5 micromachines-14-00648-f005:**
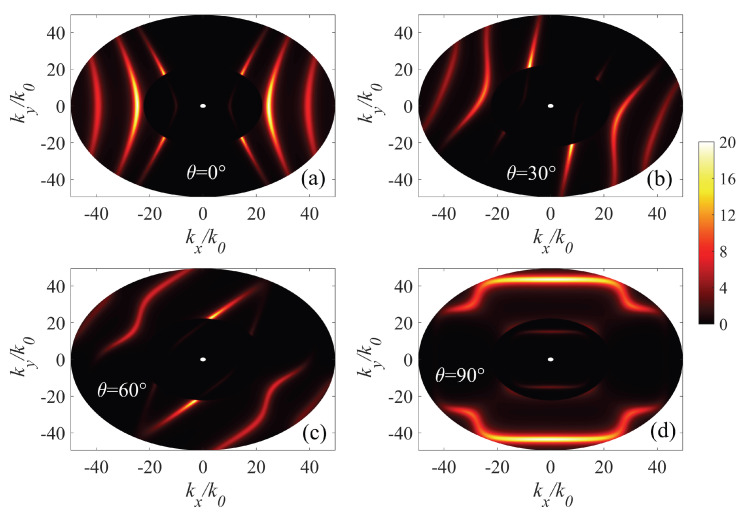
Contours of the imaginary part of the reflection coefficient ($Im(r_{pp})$) in the wave vector space at a frequency of 900 cm${}^{-1}$ under different twist angles with an external magnetic field equal to 2T. (**a**) $\theta =0^{\circ }$, (**b**) $\theta =30^{\circ }$, (**c**) $\theta =60^{\circ }$, and (**d**) $\theta =90^{\circ }$. The structural parameters are the same as used in Figure 4.

**Figure 6 micromachines-14-00648-f006:**
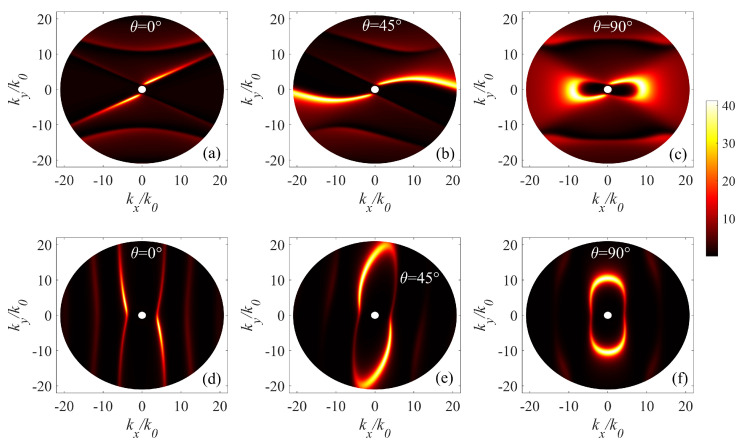
Hybridization dispersions of the α-$MoO_{3}$/α-$MoO_{3}$/InAs structure varying with wavevector components $k_{x}$ and $k_{y}$ at the wavenumber of 600 cm${}^{-1}$ (**a**–**c**), 840 cm${}^{-1}$ (**d**–**f**) with an external magnetic field equal to 2T. The thickness of top α-$MoO_{3}$ layer is 50 nm, the other parameters are the same as those used in Figure 2.

**Table 1 micromachines-14-00648-t001:** Permittivity parameters for α-$MoO_{3}$.

Direction	*x* [100]	*y* [001]	*z* [010]
$\epsilon_{\infty }$	4	5.2	2.4
$\omega_{TO}\left({\mathrm{cm}}^{-1}\right)$	972	851	1004
$\omega_{LO}\left({\mathrm{cm}}^{-1}\right)$	820	545	958
γ	4	4	2

## Data Availability

Not applicable.

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
