# Peer review of "Magnetical Manipulation of Hyperbolic Phonon Polaritons in Twisted Double-Layers of Molybdenum Trioxide"

_micromachines, 2023, doi:10.3390/mi14030648_

Round 1

Reviewer 1 Report

In this work, the authors report theoretically demonstrate a lithography-free approach for manipulating the in-plane AHPhPs in twisted double layers of α-MoO3 deposited on magneto-optic InAs substrate. The heterostructures of twisted α-MoO3 on a n-type-doped magneto-optic InAs has been proposed to realize the active manipulation of the hyperbolic phonon polaritons. This is an interesting phenomenon. As a result, I suggested the manuscript could be published in micromachines after major revisions. Some comments are listed below:

1. The phonon size should be large in several lattice parameters, why double layer of α-MoO3 can work in few layer system?

2. What is effect from the InAs substrate? Why the authors choose InAs as a substrate for magnetical manipulation? Is there others alternative candidates? 

3. Could authors supply the phonon orientation in layer or perpendicular to the layer?

4. Why are the different twist angles selected in Figures 3, 4, 5 and 6?

5. Although the active control of hyperbolic polarity can be realized in theory. What are the difficulties and challenges in his experiment? I hope the author can give some prospects.

6. There are some typos in the manuscript. The authors should spend time on improving the writing.

7. In the result, the author claimed that “Duing to the TM-polarized evanescent waves occupies the main contribution”. How TE polarized evanescent waves is reflected in the operation of AHPhP?

8. How is hyperbolic polarity characterized experimentally? Please briefly explain it.

9. There is a lack of Figure 2c. In addition, Figure 2a is the hybridization dispersion of real part while Figure 2b is the hybridization dispersion of imaginary part. How can they demonstrate the influence of different twist angles (θ = 0o and θ = 45o).

10. Figure 3 and 4 are the contours of the imaginary part of the reflection coefficient (Im(rpp)) in the wave vector space at a frequency of 900 cm1 without/with an external magnetic field. To making a distinct comparison with Figure 4, the authors should redraw Figure 3 to see if there is a topological transition.

11. The figure captions of Figure 5 and Figure 6 are same? There are lack of Figure 5e and 5f in Figure 5.

Author Response

 We thank the reviewer for the positive assessment of our work regarding the importance, novelty and potential impact.

Reviewer 2 Report

In this paper author has described a unique method to mediate the coupling of phonon polaritons in double-layers of vdW α-MoO3. The phonon polaritons profile can be changed by twisting the angle between the optical axes of the two separated layers. Effect of external magnetic field on PhPs was also shown in this study. Even though author has shown some interesting results on this system, several issues need to be addressed before publishing.

1.      The author proposed a “lithography free” geometry to control the PhPs , i.e by using two twisted MoO3. How is this experimentally feasible?

2.      How did they get the parameters in table 1? Add reference.

3.      Figure 2 caption is wrong. This is not a hybridization dispersion plot.

4.      In this study imaginary part of the reflection coefficient has been investigated. Explain why imaginary part has been investigated and what this represents?

5.      Figure 3 caption is wrong. This is not a contour plot of hybridization dispersion. Fix this.

6.      Define Kx, K0 and Ky.  

7.      Figure 4 caption might be wrong. It should be “without magnetic field”. Please check.

8.      For figure 2 plot author used 200 nm thickness for MoO3 whereas for figure 4 and 5 thickness changes to 50 nm. Any particular reason why? Also, how contour plots change with thickness.

9.      Caption for figure 5 is also wrong. This should be caption for figure 6.

1. Author mentioned magnetic field has an effect on the contour plot of the imaginary part of reflection coefficient. How does it change with different magnetic field? It would be nice to show contour plot with two different magnetic field.

Author Response

 We thank the reviewer for the positive assessment of the quality and novelty of our work.

Round 2

Reviewer 2 Report

The author has corrected the figure caption and made the changes that were asked. Some minor issues

1. Need to fix some grammatical corrections. 

2.  Comment on how to make the twisted structure experimentally